# Lottery Ticket Adaptation: Mitigating Destructive Interference in LLMs

## Abstract

Existing methods for adapting large language models (LLMs) to new tasks are not suited to multi-task adaptation because they modify all the model weights–causing destructive interference between tasks. The resulting effects, such as catastrophic forgetting of earlier tasks, make it challenging to obtain good performance on multiple tasks at the same time. To mitigate this, we propose Lottery Ticket Adaptation (LoTA), a sparse adaptation method that identifies and optimizes only a sparse subnetwork of the model. We evaluate LoTA on a wide range of challenging tasks such as instruction following, reasoning, math, and summarization. LoTA obtains better performance than full fine-tuning and low-rank adaptation (LoRA), and maintains good performance even after training on other tasks – thus, avoiding catastrophic forgetting. By extracting and fine-tuning over *lottery tickets* (or *sparse task vectors*), LoTA also enables model merging over highly dissimilar tasks.

## 1. Introduction

Large language models (LLMs) (Brown et al., 2020) have seen an explosion of applications to real-world problems (OpenAI, 2023; Team et al., 2023) via adaptation (Ouyang et al., 2022) to new tasks. Three major *multi-task adaptation* paradigms have emerged: storing and loading task-specific *adapters* (Hu et al., 2022; Beck et al., 2021), continuing to train instruction-tuned models on new tasks in serial via *sequential training* (Ouyang et al., 2022), and combining the adaptations to tasks learned in parallel via *model merging* (Ilharco et al., 2022). Each paradigm has its own associated challenges, such as catastrophic forgetting during sequential training (McCloskey & Cohen, 1989; Dong et al., 2023; Ramasesh et al., 2022; Luo et al.,

---
[1]Anonymous Institution, Anonymous City, Anonymous Region, Anonymous Country. Correspondence to: Anonymous Author <anon.email@domain.com>.

Submitted to the Workshop on Advancing Neural Network Training at International Conference on Machine Learning (WANT@ICML 2024). Do not distribute.

2023; Wang et al., 2024), and methods that have been proposed to mitigate these challenges (Crawshaw, 2020; Zhang & Yang, 2021). In this work, we propose a new LLM adaptation method, called **Lottery Ticket Adaptation (LoTA)**, that (1) provides sparse adaptation by freezing a majority of the parameters and updating only a sparse subnetwork of the base model and (2) resolves the challenges in common *multi-task adaptation* paradigms. (More details in Section 3.) We summarize our contributions:

[leftmargin=15pt]We train *lottery tickets* (or *sparse task vectors*) that can be stored efficiently and obtain performance similar to full fine-tuning (FFT) and higher than LoRA across a range of tasks spanning reasoning, math, code generation, and instruction following. When adapting Mistral for instruction following, FFT and LoTA both get a length-controlled AlpacaEval 2 winrate (Dubois et al., 2024)(how often GPT-4 prefers the outputs of our model over its own) of $19.0\%$, but LoRA only gets a winrate of $15.3\%$. We apply LoTA to mitigate **catastrophic forgetting** (McCloskey & Cohen, 1989) of earlier tasks, enabling sequential adaptation to new tasks. When adapting an instruction tuned model to a mix of new tasks, the winrate of the FFT model drops from $19.0\%$ to $0.5\%$, but by using LoTA we can limit the drop to $15.9\%$. We can use LoTA to **merge models in parallel** (Wortsman et al., 2022; Jin et al., 2022; Zhang et al., 2023a) across dramatically different tasks, achieving better performance than existing merging methods that rely on post hoc sparsification (Yadav et al., 2023) (which degrades performance for FFT models) because it naturally trains *sparse task vectors*. When merging instruction following and math models with LoTA, we get a task-average performance of $38.5\%$ where a merge of FFT models obtains $36.7\%$.

## 2. Background: Multi-Task Adaptation

In this section, we go over common multi-task adaptation paradigms and discuss the challenges existing fine-tuning methods, such as FFT and LoRA, bring in each paradigm.

**Storing and Loading Adapters.** As illustrated in the first row of Figure 1, an emerging paradigm for multi-task adaptation is to store an adapter for each desired task and load a particular adapter at inference time (Ostapenko et al.,

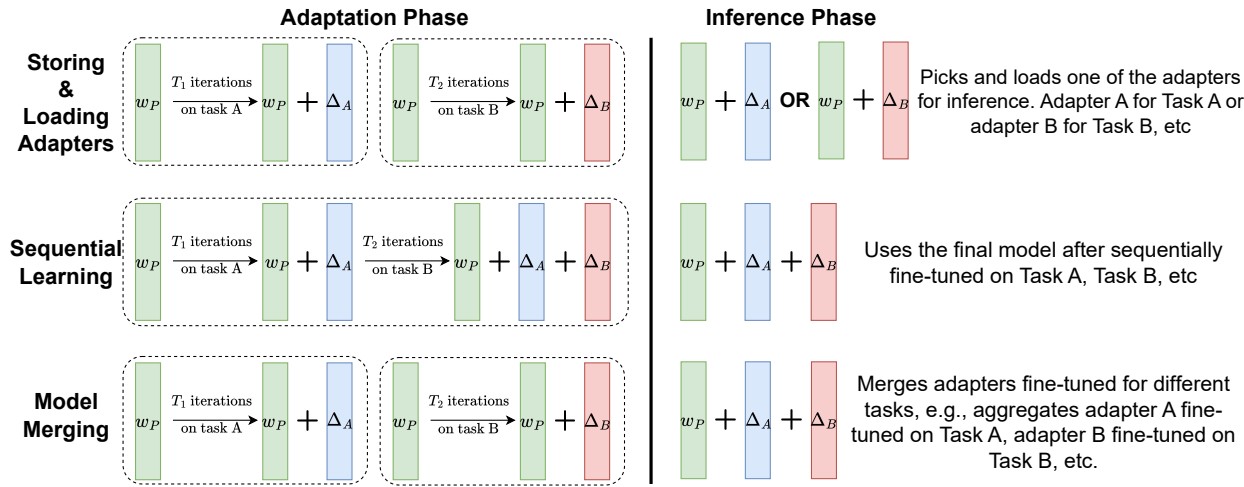

*Figure 1.* Multi-task adaptation: storing and loading adapters, sequential training, model merging.

2024; Beck et al., 2021; Mangrulkar et al., 2022), depending on the need (Houlsby et al., 2019). This approach, while avoiding any interference between tasks, increases the memory and compute cost as it requires storing and loading an additional adapter per task. To mitigate these costs, a number of PEFT methods have been developed (Zhang et al., 2023b; Li & Liang, 2021; Liu et al., 2022b; Lester et al., 2021). Among them, LoRA (Hu et al., 2022) (training adapters in the low-rank space) has received notable attention due to its simplicity. For instance, services such as Punica and S-LoRA allow developers to use this approach to serve large numbers of task-specific adapters for specific requests (Chen et al., 2023; Sheng et al., 2023). However, a persistent gap in capacity between PEFT methods and FFT has presented a tradeoff between adapter overhead and performance (Hu et al., 2022; Liu et al., 2024b; Kopiczko et al., 2023; Biderman et al., 2024; Nikdan et al., 2024).

**Sequential Training.** When it is desired to have a single model with multi-task abilities (as opposed to storing and loading adapters per task), one common approach is to fine-tune the model on different tasks sequentially (Ruder, 2017), e.g., first fine-tune on task A, then fine-tune on task B. This is summarized in the second row of Figure 1. Note that sequential training is distinct from continual pre-training, because sequential training uses the instruction tuning objective whereas continual pre-training typically uses the pre-training objective of next-word-prediction (Çağatay Yıldız et al., 2024).

Fine-tuning the LLM for new tasks with FFT or existing PEFT methods leads to catastrophic forgetting of earlier tasks. This is problematic, especially for *safety* alignment, since we can fine-tune an LLM to be safe but later get this feature erased during fine-tuning on new tasks (Lermen et al., 2023). In fact, a number of works have aimed to mitigate this vulnerability (McCloskey & Cohen, 1989; Dong et al., 2023; Ramasesh et al., 2022; Luo et al., 2023; Wang et al., 2024), leaving an open research question: *Can model trainers release aligned models that remain safe even if users fine-tune them for other, potentially malicious, tasks?*

**Model Merging.** There has been a recent interest in model arithmetic and editing methods, including merging multiple models adapted to different individual tasks to have a single model adapted to multiple tasks simultaneously (Ilharco et al., 2022). As the third row of Figure 1 shows, this is typically done via aggregating *task vectors* (or adapters) of different tasks. The existing model merging techniques either require post-processing the task vectors through sparsification (Yu et al., 2023; Davari & Belilovsky, 2023; Yadav et al., 2023), degrading the performance on the task, and/or require extensive hyperparameter tuning for a weighted aggregation of task vectors (Matena & Raffel, 2022; Xiao et al., 2023b).

## 3. Lottery Ticket Adaptation (LoTA)

Each paradigm of multi-task learning poses different challenges, and different methods have been proposed to address these challenges. We defer the more in-depth analysis of these proposed methods in Appendix A because of the sheer quantity of related work that must be covered. The number of methods itself poses challenges for studying their drawbacks, especially when these methods are adopted in settings orthogonal to those they were originally developed for (i.e., LoRA leading to catastrophic forgetting of safety alignment (Lermen et al., 2023)). *Rather than proposing tailored solutions for each paradigm of multi-task adaptation, we want to propose a simple algorithm that can serve as an effective foundation across all three paradigms*

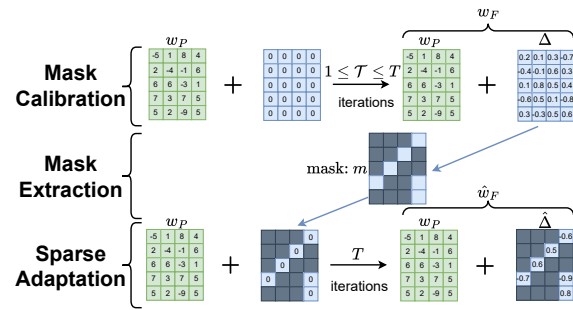

Figure 2. Lottery Ticket Adaptation (LoTA): (1) Mask calibration via FFT for $\mathcal{T}$ iterations, (2) Extracting the sparsity mask $m$ from the task vector $\Delta$, (3) Sparse fine-tuning with sparsity mask $m$ for $T$ iterations.

*and evaluate it on a wide range of challenging tasks.*

---

**Algorithm 1** Lottery Ticket Adaptation (LoTA)

**Require:** Adaptation algorithm $\mathbb{A}$, alignment dataset $\mathbb{D}$, pre-trained weights $w_P$, sparsity ratio $s$, learning rate $\eta$, number of calibration iterations $\mathcal{T}$, number of sparse training iterations $T$.

 1: **Mask Calibration:**
 2: $w_F \leftarrow w_P$
 3: **for** $\tau \in 0, \ldots, \mathcal{T}$ **do**
 4:     $\nabla = \mathbb{A}_{\mathbb{D}}(w_F)$ {Compute gradient for weights}
 5:     $w_F = w_F - \eta \cdot \nabla$ {Update the model}
 6: **end for**
 7: **Mask Extraction:**
 8: $\Delta = w_F - w_p$ {Find the task vector}
 9: $m = \texttt{Sparsify}(\Delta, s)$ {Create the sparsity mask by thresholding the task vector based on magnitude}
10: **Sparse Adaptation:**
11: $w \leftarrow w_P$
12: **for** $t \in \{0, \ldots, T\}$ **do**
13:     $\nabla = \mathbb{A}_{\mathbb{D}}(w)$ {Compute gradient for weights}
14:     $\hat{\nabla} = \nabla \odot m$ {Apply sparse mask to gradient}
15:     $w = w - \eta \cdot \hat{\nabla}$ {Update the model}
16: **end for**
17: $\hat{w}_F \leftarrow w$
**output** $\hat{w}_F$

---

The desiderata for each multi-task adaptation paradigm motivates the design of our method. For *adapters*, we want a representation that can be easily compressed for memory efficiency. For *sequential training*, we want a representation that minimizes destructive interference between the previously learned tasks and tasks to be learned in the future. For *model merging*, we want representations that are mutually sparse with each other in parameter space to again prevent destructive interference. We now propose LoTA, a single method that enjoys all these features.

We first describe the workflow of LoTA, then revisit the problems each multi-task paradigm faces and discuss how and why LoTA successfully mitigates them.

**Lottery Ticket Adaptation (LoTA).** LoTA works in two phases as summarized in Figure 2: (1) mask calibration, (2) mask extraction, (3) sparse adaptation. In the mask calibration phase of LoTA, a base model with parameters $w_P$ is fine-tuned for $\mathcal{T}$ iterations, yielding a fine-tuned model with parameters $w_F$. Then, in the mask extraction phase, LoTA extracts a sparsity mask from the task vector $\Delta = w_F - w_P$ based on the magnitude of the updates in $\Delta$. $\mathcal{T}$ could be as small as one iteration. In the sparse adaptation phase of LoTA, the model is first reset to its original state with weights $w_P$. Then the subnetwork $w_P \odot m$ is fine-tuned for $T$ iterations, while leaving the remaining parameters $w_P \odot (1 - m)$ frozen at their initial values. We summarize the workflow of LoTA in Figure 2 and Algorithm 1 further.

By confining the adaptation updates within subnetworks (identified by $m$), **LoTA is able to mitigate destructive interference**, e.g., adaptation loss during fine-tuning on future datasets or model merging, that FFT and LoRA suffer from. We discuss this in more detail under three multi-task adaptation paradigms below and provide empirical comparisons with FFT and LoRA in Section 5.

**(1) Storing & Loading Adapters.** As mentioned before, PEFT methods have emerged to reduce the memory cost of storing and loading adapters. However, the most popular and commonly used PEFT method, LoRA, restricts the adaptation updates to have a low rank, which does not capture the complex downstream tasks (Nikdan et al., 2024). In parallel, recent work (Isik et al., 2023) on compressing the delta between the fine-tuned and pre-trained model $\Delta = w_F - w_P$ suggests that FFT updates are highly compressible through a simple magnitude-based sparsification. LoTA exploits this underlying sparsity during fine-tuning and obtains better performance than LoRA, while requiring fewer parameters to be trained.

**(2) Sequential Training.** Existing fine-tuning methods, such as FFT and LoRA, are known to cause catastrophic forgetting (Lermen et al., 2023). This is particularly concerning for safety alignment as any safety measure the model developers add could be erased by further fine-tuning. By restricting the task vectors to be sparse, LoTA provides robustness against catastrophic forgetting–improving the durability of previous alignments. As LoTA prevents destructive interference between sequential tasks via sparse and disjoint task vectors, the same phenomenon is also helpful in adapting to new tasks. To further enhance the robustness against destructive interference in sequential training, we propose **Lottery Ticket Together Optimization (LoTTO)** which learns mutually sparse (i.e.,

non-overlapping) masks for sequentially learned tasks.

**Further Enhancing Sequential Training via Lottery Ticket Together Optimization (LoTTO)** Without loss of generality, suppose we have two tasks, Task A and Task B, and that we have already learned Task A with LoTA. LoTTO calibrates a sparsity mask for Task B by first training a model where the only weights that can be updated are those that are *not* updated when running LoTA on Task A, and then using a sparse set of those weights to train the final model. This procedure can be applied inductively to enable sequential adaptation to multiple tasks, so that a model developer seeking to adapt a model adapted with LoTA (potentially on several tasks), just needs to ensure that they do not update the task vector with respect to the base model.

**(3) Model Merging.** Existing model merging methods typically aim to merge task vectors of relatively similar language datasets (Wortsman et al., 2022; Jin et al., 2022; Zhang et al., 2023a), and they do so after a post hoc sparsification (Yadav et al., 2023; Yu et al., 2023; Davari & Belilovsky, 2023) of the task vectors. The post hoc sparsification aids in ensuring the task vectors are disjoint – hence can prevent destructive interference – but, in return, degrades the performance of each individual task. LoTA, on the other hand, enforces sparsity during fine-tuning and directly trains *sparse task vectors*, obviating the need for post hoc sparsification.

## 4. Experimental Setup

In this section, we provide details of the experimental setup, including the baselines, model, dataset, and metric selection. We present the results later in Section 5. We are limited to an academic computing budget, and all results are conducted with a single A100 GPU. We typically do $1 - 3$ epochs of training for each dataset as this is a standard choice in LLM fine-tuning and indicate it specifically for single-epoch fine-tuning. We use the RMSProp optimizer with default hyperparameters.

**Baselines & Hyperparameters.** Across all three multi-task adaptation paradigms, we compare LoTA against FFT and LoRA. We extensively tune the hyperparameters for FFT and LoRA to ensure that we are comparing against strong baselines. We do not tune hyperparameters for LoTA and directly transfer the hyperparameters from FFT. We fix the sparsity ratio hyperparameter in LoTA to $90\%$, so that the number of parameters updated roughly matches that of the best-performing LoRA rank of 256. We provide an ablation study with higher sparsity levels in Section 5.4. We report all ranges for hyperparameters in Appendix B.

**Models.** We use the best performing open-weights model families, Mistral (Jiang et al., 2023) and Llama 3 (AI@Meta, 2024; Touvron et al., 2023), specifically

Mistral-7B and Llama-3-8B – the largest models we can adapt with FFT on a single GPU.

**Tasks** We consider six capabilities: instruction following, safety, math, coding, summarization, and reasoning. We now briefly discuss each capability, the datasets we use to fine-tune and evaluate the presented methods, and the motivation behind the choices.

**Instruction Following.** The most widely-used instruction-tuned LLMs are the "Instruct" or "chat" versions of base models, such as Llama-3-8B-Instruct (AI@Meta, 2024). This is because the process of tuning models on human instructions aligns models to human preferences across a range of tasks (Ouyang et al., 2022). For this, we adapt models to data from UltraFeedback (Cui et al., 2023), which contains a mixture of datasets covering truthfulness, honesty, and helpfulness in addition to instruction-following. We measure the instruction following ability by length-controlled AlpacaEval2 Win Rate (Li et al., 2023), which we refer to as "winrate". A high winrate means that GPT-4 (OpenAI, 2023) prefers the responses of our model on a set of representative prompts over its own responses. Winrate is the metric most closely correlated with human rating preference (Dubois et al., 2024). Another common benchmark for "chat" models is MT-Bench (Zheng et al., 2023), but there is a significant degree of data contamination between MT-Bench and other task-specific training datasets (Yu et al., 2024)–hence, we do not evaluate on MT-Bench.

**Reasoning.** We train on the standard set of 8 commonsense reasoning tasks (Christopher et al., 2019; Bisk et al., 2019; Sap et al., 2019; Zellers et al., 2019; ai2, 2019; Clark et al., 2018; Mihaylov et al., 2018) (Boolq, PIQA, SocialIQA, Hellaswag, Winograde, ARC-easy, ARC-challenge, OpenBookQA) and report the exact-match accuracy on the test set. As a representative task, we use ARC-easy.

**Math.** We use the set of 9 math instruction datasets from (Yue et al., 2023) for fine-tuning and report performance on the test set of GSM8k (Cobbe et al., 2021). When only considering a single task, we choose GSM8k as the representative task as it is commonly used as a single training and test task by other papers.

**Code Generation.** We use data that instructs the model to write SQL queries given some context (b mc2, 2023)(SQL-create-context) and report the ROUGE-1 F1 score (Lin, 2004) on the test set.

**Summarization.** We use data from Samsum (Gliwa et al., 2019) and report the ROUGE-1 F1 score on the test set.

**Safety.** We define safety as a latent capability generated by instruction tuning. A recent concern in AI policy is that, while frontier models such as GPT-3.5/GPT-4 are aligned, they can also be fine-tuned and this presents an opportunity

to misalign them. Recently, Qi et al. (2023) show that by fine-tuning GPT-3.5 on just 100 harmful examples for a few epochs, they can ask it to answer harmful queries that it ordinarily would refuse, and Zhan et al. (2024) show the same for GPT-4. Lermen et al. (2023) show that this can be done with LoRA rather than the fine-tuning method OpenAI are using in their fine-tuning API (presumably FFT). This is more than an academic concern; SB-1047 (Scott Weiner, 2024), recently passed in California, requires model developers providing access to frontier models to implement best effort mitigations to prevent users from misaligning those models. We evaluate the safety of our models on HEx-Phi (Qi et al., 2023), a dataset of 330 questions spanning multiple categories such as malware, fraud, etc. The safety score (higher is better) is the percentage of queries from the test set where the model refuses to respond. Aligned models such as Llama-3-Instruct will score $100\%$ on this task, but because we are doing the alignment ourselves starting from a base model, our baseline Instruct model only gets $93\%$ on this task. Given that we are primarily interested in measuring the *forgetting* of safety alignment, we do not see this as a major limitation.

## 5. Experimental Results

We evaluate LoTA and the baseline methods, FFT and LoRA, across all three paradigms of multi-task adaptation. In all experiments, we use LoTA with $90\%$ sparsity where the mask is calibrated by training for *a single epoch* ($\mathcal{T}$ is one epoch) on the adaptation dataset. We choose $90\%$ sparsity because this is a nontrivial sparsity level that we find achieves good performance across the range of tasks we consider. We ablate the level of sparsity and amount of calibration data in Section 5.4.

### 5.1. Adapting to a Single Task

We first consider the simplest setting, in which we fit an adapter to each dataset of interest starting from a pre-trained base model, i.e., one adapter per task. In Table 1, we find that LoTA outperforms LoRA and performs similarly to FFT. Although LoRA is able to achieve similar performance to LoTA on the easier tasks, such as SQL, Samsum, there is a clear gap in performance on the more challenging tasks, such as Instruction Following and GSM8k. LoTA consistently recovers the performance of FFT.

LoRA has a heavy regularizing effect on training. In some settings, regularization can improve performance. On the commonsense reasoning task, because the base model can get nontrivial performance via in-context learning (although we never use in-context learning in our evaluations), regularizing the training as LoRA does actually improves the score on reasoning for Mistral. Note here that we do a grid search over learning rate and rank for LoRA, and the best performance is at $r = 256, 1e-5$. However, sparsity also

has a regularizing effect, and LoTA is even more successful on the reasoning task when using the same learning rate as we searched for the base model ($1e-6$).

In other settings, e.g., GSM8k, the regularization hurts performance significantly; (Nikdan et al., 2024) report a similar-sized gap between LoRA and FFT on this dataset when training LLama-2-7B. Because LoRA is underfitting the data, it may be better suited for settings where we only train for a single epoch, or on smaller datasets. In Table 6 in Appendix B, we make a side-by-side comparison of LoRA and LoTA when training for a single epoch and find that LoTA outperforms LoRA significantly across all tasks; in fact, the difference is even more pronounced after a single epoch.

**Storing LoTA Adapters Efficiently.** Although FFT generally performs better than PEFT methods, it is typically infeasible to store a full copy of the model weights for each task. Practitioners, therefore, consider a tradeoff between memory and adaptation performance when loading task-specific adapters. We now discuss the memory consumption of LoTA and LoRA. Storing the sparse task vector from LoTA requires $64$ bits per parameter, $32$ from the parameter, and $32$ for the metadata needed to store the location of the parameter. The latter breaks down as 5 bits for the layer index, 3 bits for the module index within the layer, and 12 bits for each of the layer input and output dimensions in delta encoding. If we use $90\%$-sparse LoTA, our task vector is compressed $5\times$; if we use $99\%$-sparse LoTA, our task vector is compressed $50\times$. The memory-utility tradeoff between LoTA and LoRA can be quantified in terms of each method's performance at a given level of compression, which translates into the sparsity level for LoTA and the rank $r$ for LoRA. We provide a further comparison in Section 5.4 as ablation.

### 5.2. Sequential Training

During sequential training, a model is first adapted to one capability (Task A) and then to another capability (Task B) (such as when creating a specialized model for math) or a set of capabilities (the most common setting for open-sourced fine-tunes of frontier models). The main challenges we seek to mitigate are (1) catastrophic forgetting of Task A and (2) the inability to adapt to Task B. We set instruction following as Task A in all settings because this maps to the enterprise adaptation setting, where user queries need to be answered with a combination of instruction following and domain-specific knowledge. In particular, OpenAI offers a fine-tuning API for their aligned models (GPT-3.5 and GPT-4) (OpenAI, 2023).

**Mitigating Catastrophic Forgetting While Enabling Adaptation to Downstream Datasets.** In Table 2 we consider the a range of method combinations for a simplified setting where we seek to adapt an Instruct model

*Table 1.* Performance comparison on single-task datasets for 3 epochs if sparse adaptation. We report the winrate on instruction following, the accuracy of exact match on the reasoning and math tasks, and the ROUGE-1 score on the SQL generation and summarization tasks. LoTA outperforms LoRA on the challenging tasks of instruction following, reasoning, and math, obtaining comparable performance to FFT. **bold**: best method, underline: second best method.

| Model | Method | Instruction Following | Reasoning | GSM8k | SQL | Summarization |
|-------|--------|----------------------|-----------|-------|-----|---------------|
| 3*Mistral | FFT | $\underline{19.0_{0.98}}$ | 83.5 | $\mathbf{59.8_{1.0}}$ | $98.9_{0.1}$ | $52.0_{0.2}$ |
| | LoRA | $15.3_{0.8}$ | $\underline{85.4}$ | $54.3_{1.1}$ | $98.9_{0.1}$ | $52.9_{0.3}$ |
| | LoTA (ours) | $\mathbf{19.0_{0.7}}$ | $\mathbf{87.0}$ | $\underline{59.1_{1.1}}$ | $\mathbf{98.9_{0.1}}$ | $\mathbf{52.9_{0.2}}$ |
| 3*Llama 3 | FFT | $\underline{17.61_{0.8}}$ | $\mathbf{84.8}$ | $\mathbf{63.0_{0.1}}$ | $\mathbf{99.4_{0.1}}$ | $\mathbf{53.6_{1.9}}$ |
| | LoRA | $14.2_{0.8}$ | 82.2 | $54.7_{0.4}$ | $98.7_{0.1}$ | $52.3_{0.2}$ |
| | LoTA (ours) | $\mathbf{18.0_{0.7}}$ | $\underline{84.1}$ | $\underline{61.8_{0.7}}$ | $\underline{99.0_{0.1}}$ | $\underline{52.3_{0.3}}$ |

*Table 2.* Sequential learning first on Task A (Instruction Following) then on Task B (varied). Fine-tuning on Tasks A and B is performed using both FFT and LoTA. The utility of each task is computed after fine-tuning on Task B is completed. Note that there is no utility when we fine-tune on harmful data to evaluate the catastrophic forgetting of Safety, and the baseline is the safety score of the Instruct model. FT=Fine-tuning. We reproduce the baseline of doing FFT on each method independently as reported in Table 1 for convenience; note that on the reasoning task LoTA outperforms FFT. **bold**: best method, underline second-best method; we do not report second-best when only two methods are presented. All results are with Mistral. We reuse the same mask for LoTTO calibrated on GSM8k for MathInstruct, Reasoning, GSM8k+Arc+SQL and Safety.

| Task B | Method on Task A | Method on Task B | Utility of Task A (Drop) | Utility of Task B (Drop) |
|--------|-----------------|-----------------|--------------------------|--------------------------|
| Instruction Following | Baseline | - | 19.0 (-) | - |
| 7*GSM8k | - | Baseline | - | 59.8 (-) |
| | FFT | FFT | 15.2 (3.8) | 58.3 (1.5) |
| | LoTA (ours) | FFT | $\underline{17.7}$ (1.3) | $\underline{58.7}$ (1.1) |
| | FFT | LoTA (ours) | 15.9 (3.1) | 54.2 (5.6) |
| | LoTA (ours) | LoTTO (ours) | $\mathbf{17.8}$ **(1.2)** | $\mathbf{59.1}$ **(0.7)** |
| | FFT | LoRA | 14.1 (4.2) | 55.5 (4.9) |
| | FFT | FFT (Mixed) | 16.3 (2.7) | 55.5 (4.3) |
| 2*MathInstruct | - | Baseline | - | 56.7 (-) |
| | FFT | FFT | $14.2_{0.8}$ (4.8) | $51.3_{0.2}$ (5.4) |
| | LoTA (ours) | LoTA (ours) | $\mathbf{16.0_{0.7}}$ $(-3.0)$ | $\mathbf{55.5_{0.1}}$ (1.2) |
| 3*Reasoning | - | Baseline | - | 83.5 (-) |
| | FFT | FFT | $0.2_{0.1}$ (18.8) | 82.3 (1.2) |
| | LoTA (ours) | LoTTO (ours) | $\mathbf{16.5_{0.9}}$ (2.5) | $\mathbf{83.7}$ (-) |
| 4*GSM8k+Arc+SQL | - | Baseline | - | 77.0 |
| | FFT | FFT | $0.5_{0.2}$ (18.6) | 75.0 (2.0) |
| | LoTA (ours) | FFT | $11.5_{0.7}$ (7.5) | $\mathbf{75.4}$ **(1.6)** |
| | LoTA (ours) | LoTTO (ours) | $\mathbf{15.9_{0.9}}$ (3.1) | 73.8 (3.2) |
| 3*Safety | Baseline | - | 93.1 (-) | - |
| | FFT | FFT | $19.1_{3.5}$ (73.9) | - |
| | LoTA (ours) | LoTTO (ours) | $\mathbf{63.4_{2.2}}$ (29.7) | - |

to Math data without catastrophic forgetting. We will go row-by-row through the table and analyze each set of results. Even when training on just a single, relatively small dataset, FFT in both phases suffers a significant drop in winrate. An easy way to mitigate this is to simply train the initial Instruct model with LoTA. Following this, FFT on GSM8k does not significantly reduce winrate. However, it does present a potentially unwelcome tradeoff in task accuracy. For this, we turn to LoTTO, which achieves the best performance across both tasks.

**Does LoRA Really Forget Less, or Does It Just Learn Less?** Recent work evaluates the performance of pre-trained models before and after fine-tuning on domain-specific tasks with LoRA (Biderman et al., 2024; Ghosh et al., 2024) and concludes that LoRA is less prone to catastrophic forgetting than FFT. In Table 2 we find that adapting the FFT Instruct model with LoRA does not lead to less degradation in

winrate than adapting the FFT Instruct model with FFT, but it does lead to worse performance on the downstream task.

**How Much Does Data Reuse Help?** The simplest and arguably most performant method from prior work that we found for mitigating catastrophic forgetting is simply to mix in some in-distribution data from Task A. This is the line marked with "FFT (Mixed)", and it does mitigate forgetting on Task A, but at the cost of performance on Task B. We ablate the amount of data from Task A to be used between $1 - 100\%$ of the dataset size of the data from Task B, and this is the best result in terms of mitigating forgetting on Task A. As we mix in less and less data from Task A, this method approaches just doing FFT sequentially in performance, so we omit those results for brevity.

LoTTO adds an additional layer of computational overhead and is somewhat impractical; if we need an additional

level of calibration for each task, we can run out of parameters to update quite fast. However, we find that the mask we calibrate for GSM8k can be used even when other data is present. We now consider the more challenging setting when we need to adapt to *multiple* tasks without catastrophic forgetting while still obtaining good performance on those tasks.

In Table 2 we adapt an Instruct model to a mix of reasoning, math, and SQL data and find a surprising result; the FFT Instruct model collapses almost completely to a winrate of less than half a percent. As we showed in Table 2 this can be mitigated by mixing in more instruction following data, albeit at a cost. The LoTA Instruct model still degrades in performance ($19 \rightarrow 11$) but nowhere near as much. In the line marked "LoTTO", we instead adapt the LoTA Instruct model to the downstream tasks using the mask calibrated from GSM8k with LoTTO. Applying LoTTO to make the downstream adaptation be mutually sparse with the initial LoTA Instruct model increases performance significantly on instruction following and math. Performance on Arc and SQL suffers somewhat because our mask does not consider those tasks, but this means that our mask is much cheaper to calibrate and is, in fact, generalizable not only across tasks within a domain (as shown in Table 2) but also *across* domains.

**Mitigating Catastrophic Forgetting of Safety Alignment**

In the "Safety" row of Table 2, we consider fine-tuning the Mistral Instruct model we trained ourselves on the 100 harmful instructions from (Qi et al., 2023). The baseline model gets a score of $80\%$ and training with FFT quickly degrades safety. Adapting the LoTA Instruct with LoTTO (again, with the LoTTO mask calibrated from GSM8k) mitigates this safety drop significantly, even though our LoTTO mask was calibrated on an extremely different dataset. Therefore, a potential mitigation would be for an entity providing a fine-tuning API such as OpenAI to do the safety training with LoTA, calibrate the LoTTO mask on a utility dataset, and then do fine-tuning on their client's dataset with LoTTO.

We do not intend to present our method as an active defense against fine-tuning attacks; given sufficient data and access to the model weights, any attacker can of course undo safety tuning entirely. However, catastrophic forgetting of safety alignment is an important problem with real-world applications, and we find it compelling that our method can mitigate this.

### 5.3. LoTA for Model Merging

We now consider the setting of model merging, where we train models on disjoint datasets fully in parallel and then merge together the task vectors with the goal of producing a model with good performance on multiple tasks. Prior

work in model merging mostly considers merging similar datasets, such as the commonsense reasoning datasets, but it is relatively easy to merge models when the datasets are similar and becomes increasingly hard as the datasets become more heterogeneous due to the gradient mismatch (Daheim et al., 2024).

**Merging Models.** We use TIES-Merging (Yadav et al., 2023) to merge the Instruct and Math models together. TIES performs post-hoc sparsification on each task vector and requires a 2-D hyperparameter search for this quantity, which we perform for the merge of FFT models. Naturally, we could optimize the performance of Task A by fully sparsifying Task B, and vice versa; we report the result that achieves good performance on Task B while maintaining some performance on Task A, and report the full range of hyperparameters in Appendix B. LoTA is inherently sparse, so when we merge a LoTA model with an FFT model we do not need to perform hyperparameter search on the LoTA model, and we use the same level of sparsity for the FFT model that we obtained when merging together two FFT models. When merging two LoTA models together, no hyperparameter search is required at all as both models are inherently sparse. We could in theory sparsify the LoTA models beyond their existing levels with post-hoc sparsification, but we do not tune this hyperparameter.

**Challenge of Overlapping Sparsity in Model Merging.** In the sequential training paradigm, we exploited the fact that masks for different tasks have a significant overlap in order to generalize our LoTTO mask calibrated on GSM8k to provide robustness to forgetting across a range of other tasks. However, this same phenomenon of overlapping sparsity presents a challenge in the model merging setting. The challenge is that because the merging is parallel, we cannot use LoTTO to calibrate the masks to be disjointly sparse as we did in the sequential training setting.

In Table 3 we merge together models trained on GSM8k and Samsum with the Mistral model that we trained for instruction following. We consider the full combination of merging FFT and LoTA models. The merge of two FFT models performs poorly in all settings on both tasks, indicating that post-hoc sparsification does not perform well for heterogeneous tasks, which is in line with recent model merging theory (Daheim et al., 2024). The merges that contain LoTA models have better performance across all tasks, but there is no combination that is pareto-optimal across all tasks.

### 5.4. Ablations

**Sparsity.** We vary the sparsity parameter in LoTA in Table 4. Multiple sparsity thresholds work well; a small amount of sparsity (i.e., $10\%$) seems to have a negative impact on the model, but this is within the error bars and may just be variance. Attempting to achieve $99\%$ sparsity from the task

*Table 3.* Model merging of Task A, Instruction Following, and Task B, (varied). Fine-tuning on Tasks A and B is performed using both full fine-tuning (FFT) and LoTA. The utility of each task is computed after merging the two task vectors. **bold**: best result, underline: second best result. We reproduce the baseline for each task on FFT from Table 1 for convenience; note again that for some tasks the LoTA baseline outperforms the FFT baseline. All results are with Mistral.

| Task B | Method on Task A | Method on Task B | Utility of Task A (Drop) | Utility of Task B (Drop) |
|---|---|---|---|---|
| Instruction Following | Baseline | - | 19.0 (-) | - |
| 4*GSM8k | - | Baseline | - | 59.8 (-) |
| | FFT | FFT | $6.9_{0.5}$ (12.1) | $59.1_{0.1}$ (0.7) |
| | LoTA (ours) | FFT | $\mathbf{16.1}_{0.8}$ (2.9) | $\mathbf{60.5}_{1.1}$ (+0.7) |
| | FFT | LoTA (ours) | $14.0_{0.8}$ (5.0) | $\underline{59.3}_{1.0}$ (0.5) |
| | LoTA (ours) | LoTA (ours) | $\underline{15.1}_{0.8}$ (3.9) | $58.4_{1.1}$ (1.5) |
| 5*Samsum | - | Baseline | - | 52.0 (-) |
| | FFT | FFT | $8.1_{0.7}$ (10.9) | $47.0_{0.0}$ (5.0) |
| | LoTA (ours) | FFT | $\underline{13.8}_{0.8}$ (5.2) | $51.8_{0.2}$ (0.2) |
| | FFT | LoTA (ours) | $10.4_{0.7}$ (8.6) | $\mathbf{53.3}_{0.1}$ (+1.3) |
| | LoTA (ours) | LoTA (ours) | $\mathbf{15.4}_{0.9}$ (3.6) | $\underline{52.7}_{0.1}$ (0.1) |
| 4*GSM8k+Samsum | - | Baseline | - | 55.9 (-) |
| | FFT | FFT | $7.6_{0.6}$ (11.4) | $49.7_{1.1}$ (6.2) |
| | LoTA (ours) | FFT | $8.9_{0.6}$ (10.1) | $\underline{50.7}_{1.1}$ (5.2) |
| | FFT | LoTA (ours) | $\underline{10.7}_{0.7}$ (8.3) | $\mathbf{55.0}_{1.1}$ (0.9) |
| | LoTA (ours) | LoTA (ours) | $\mathbf{13.1}_{0.8}$ (5.9) | $46.6_{1.0}$ (9.3) |

*Table 4.* The impact of varying the sparsity ratio on the performance of LoTA. 99%* denotes the model calibrated with iterative LoTA.

| Sparsity | 0% | 10% | 25% | 50% | 75% | 90% | 99% | 99%* |
|---|---|---|---|---|---|---|---|---|
| Performance | $19.0_{1.0}$ | $18.2_{1.0}$ | $18.5_{1.0}$ | $18.2_{1.0}$ | $19.5_{0.9}$ | $19.0_{1.0}$ | $13.1_{0.8}$ | $17.4_{0.9}$ |

vector of an FFT model does not yield good results. Instead, we can first obtain a mask with 90% sparsity, then train a LoTA model with this mask, and then use the task vector from the 90%-sparse LoTA model to calibrate the mask for our 99%-sparsity model, which we denote as 99%*. Composing multiple steps of calibration trades off performance in the high-compression setting with compute overhead.

**Compute Overhead.** Arguably, the main reason why PEFT methods such as LoRA are commonly used is because they reduce the memory consumption in the backward pass, thus enabling the use of a larger pass. (Biderman et al., 2024) recently critically analyzed this and found that to fit tasks such as math and code generation, LoRA needs to use high ranks, which in turn reduces the speedup to at most 15%. We nevertheless acknowledge that any and all PEFT methods will be faster than LoTA during *training* because LoTA requires an initial pass over the dataset to calibrate the sparsity mask.

*Table 5.* LoTA performance degrades gracefully as a smaller fraction of the data is used on the most challenging task (instruction following). 0 data usage = just creating a random mask.

| Data Used | 10% | 1% | 0 |
|---|---|---|---|
| Performance Drop (from 100%) | 4.2% | 10% | 20% |

In Table 5, we now consider how the performance drops if we calibrate the mask for a fraction of the overall dataset, including the baseline where we use a random mask which corresponds to 0% data used. LoTA can be efficiently calibrated, and even training on a small fraction of the dataset can provide 90% of the performance of the fully calibrated mask. Furthermore, instruction tuning datasets are generally quite small (we completed all experiments

in a few hours on just a single GPU). Given that the mask can be efficiently calibrated, transferred from other datasets, or in the worst case, calibrated on the entire dataset in a few hours, we do not anticipate that the compute overhead of LoTA will be a major limitation of the method.

**Storage Cost of Adapters.** Comparing the compression-utility tradeoff between LoRA and LoTA is challenging because the size of the saved LoRA adapter is determined by the rank parameter $r$, and more is not always better (which is why we have to tune $r$ for every task for LoRA). As a single point of comparison, we can look at the performance on instruction following, where 99%-sparse LoTA (17.4) outperforms LoRA for any rank (best performance of 15.3 achieved at $r = 64$, increasing rank reduces performance), and they both achieve a compression factor of $\approx 50\times$. For levels of compression beyond $100\times$, i.e., LoRA $r = 8$, LoTA does not perform well.

## 6. Discussion

We propose Lottery Ticket Adaptation (LoTA), a sparse alignment framework that fine-tunes only a sparse subnetwork of the base model, leaving the rest of the parameters frozen. LoTA successfully mitigates destructive interference (a problem with existing fine-tuning methods including full fine-tuning and low-rank adaptation (LoRA)) in many multi-task adaptation paradigms, prevents catastrophic forgetting of earlier tasks, including safety, and allows for successful model merging of even dramatically different tasks. Due to the page limit, we discuss related work in Appendix A.

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

# A. Related Work

**Model Pruning & Quantization**    Model pruning and quantization have been receiving increased attention for efficient storage and/or inference of large models. While most of the existing methods prune (Frantar & Alistarh, 2023; Dettmers et al., 2023b; Kim et al., 2023) or quantize (Frantar et al., 2023; Dettmers et al., 2022; Xiao et al., 2023a; Lin et al., 2023) the model weight directly, some focus specifically on compressing the *task vectors* (Isik et al., 2023; Liu et al., 2024a; Yao & Klimovic, 2023) through post-training sparsification or quantization, assuming that the base model is worth the storage cost since it is being used frequently for many tasks. Our proposed PEFT method, LoTA, builds on this observation that the task vectors are highly compressible through sparsification and imposes this sparsity constraint at the beginning of fine-tuning to train *sparse task vectors*.

**Lottery Ticket Hypothesis**    Motivated by the success of pruning methods at extreme sparsity ratios (Han et al., 2015), (Frankle & Carbin, 2019) proposed the lottery ticket hypothesis (LTH), claiming the existence of sparse subnetworks (or lottery tickets) that could be trained from scratch to a performance comparably to training the dense model from scratch. While this could potentially provide a way to train models sparsely more efficiently rather than training them densely and pruning them later, finding the lottery tickets, i.e., the sparsity masks, is costly. Initially, (Frankle & Carbin, 2019) proposed first training the models densely and then extracting the sparsity mask based on the magnitude of the trained dense model's weights. Later, a number of more efficient methods were proposed to find the sparsity masks more efficiently, earlier in the dense training stage (Frankle et al., 2021; Lee et al., 2019; Wang et al., 2020; Tanaka et al., 2020). Our work shows that LTH works successfully for fine-tuning LLMs as well–giving us a sparse adaptation tool, LoTA. We extensively study the tradeoff between the cost of finding the sparsity masks and the performance of the sparsely fine-tuned model. Unlike other studies on LTH for LLM adaptation (Yuan et al., 2024; Xu & Zhang, 2024), our main focus and motivation is to mitigate destructive interference in multi-task adaptation.

**Parameter-Efficient Fine-Tuning (PEFT)**    Many practitioners fine-tune already pre-trained LLMs with less data and compute instead of training them from scratch (Liu et al., 2022a; Wang et al., 2022; Ouyang et al., 2022). While this reduces the cost of LLM training significantly, fine-tuning each and every parameter of these large models for each (or a few) task is still very costly. This has led to a number of parameter-efficient fine-tuning (PEFT) methods reducing the number of trainable parameters during fine-tuning (Zhang et al., 2023b; Li & Liang, 2021; Liu et al., 2022b; Lester et al., 2021; Edalati et al., 2022; Hu et al., 2023; Nikdan et al., 2024; Guo et al., 2020). Among different PEFT methods, low-rank adaptation (LoRA) (Hu et al., 2022) and its variants (Dettmers et al., 2023a; Guo et al., 2024; Li et al., 2024; Kopiczko et al., 2024) have shown similar performance to full fine-tuning in many tasks while reducing the number of trainable parameters through low-rank approximation to model updates during fine-tuning. Our PEFT method, LoTA, while reducing the number of trainable parameters significantly via sparsity, has various other benefits in different applications, such as avoiding catastrophic forgetting (of especially safety alignment), enabling fine-tuning on new tasks more successfully, model merging using sparse task vectors, unlearning, and communication-efficient federated learning (FL). We demonstrate that full fine-tuning and the existing PEFT methods fall short in these applications and significantly underperform LoTA.

**Catastrophic Forgetting**    When LLMs go through sequential (or continual) multitask learning, i.e., fine-tuned on different tasks sequentially, they often suffer from performance loss on earlier tasks–known as catastrophic forgetting (McCloskey & Cohen, 1989; Dong et al., 2023; Ramasesh et al., 2022; Luo et al., 2023; Wang et al., 2024). To mitigate this, a number of data-centric and architectural solutions have been proposed for language and other domains. Replay-based methods (Rebuffi et al., 2017; Romanov et al., 2018) add a portion of the previously learned data during fine-tuning on a new task, which raises privacy concerns as it requires constant access to previously learned data. Regularization-based approaches (Huang et al., 2021; Aljundi et al., 2018) tend to have poor adaptability to specific tasks. An architecture-based approach, "progressive prompts" (Razdaibiedina et al., 2023), sequentially concatenates soft prompts as they are being learned for each task–showing some resistance against forgetting. However, they require access to task identifiers at inference for each task, which is not always feasible. Other architecture-based approaches add additional modules to learn task-specific abilities (Dou et al., 2023; Wu et al., 2024)–requiring customized deployment due to architecture change. Closest to our work, (Hui et al., 2024) updates a randomly selected subset of the parameters at each iteration of fine-tuning to preserve the earlier tasks in the not-updated parameters of that iteration. Despite similarities, our work LoTA (1) uses a fixed sparsity mask throughout fine-tuning instead of a new mask at every iteration, which yields sparse task vectors that are useful for other applications such as model merging and communication-efficient FL, and (2) finds data-dependent masks rather than the randomly selected masks in (Hui et al., 2024). Furthermore, unlike (Hui et al., 2024), LoTA not only preserves the earlier tasks on

frozen parameters but also constrains the new tasks on a highly sparse subnetwork–providing resistance to catastrophic forgetting even when malicious users attempt to overwrite the earlier tasks via FFT.

**Model Merging**    Merging multiple task-specific models into a single model with multitask abilities (Wortsman et al., 2022; Jin et al., 2022; Zhang et al., 2023a) has been an appealing alternative to sequential multitask learning, which suffers from catastrophic forgetting and could be inefficient, especially if task vectors are already available. The existing model merging methods include averaging weights of task-specific models (Wortsman et al., 2022), task arithmetic through combining task vectors (Ilharco et al., 2022), weighted aggregation of parameters (Matena & Raffel, 2022; Xiao et al., 2023b), combining task vectors after some post-processing such as trimming low-magnitude deltas (Yadav et al., 2023) or sparsifying the deltas (Yu et al., 2023; Davari & Belilovsky, 2023). Our method, LoTA, directly learns *sparse task vectors*, obviating the need to post-process the task vectors, and outperforms existing model merging methods. Most importantly, LoTA enables merging task vectors trained on heterogeneous datasets, while the other model merging methods are often limited to similar datasets. This advancement is an important step towards scalable FL (Kairouz et al., 2021) with LLMs as it enables merging *sparse task vectors*, which brings communication efficiency, trained over heterogeneous datasets (of each edge device).

We note that we test model merging with LoTA specifically for highly dissimilar datasets to show its compatibility with FL, which considers edge devices with heterogeneous datasets. When used in FL, LoTA can reduce communication and memory costs significantly, which is a main bottleneck when scaling FL to large models. The successful use of LoTA for model merging and arithmetic further shows its promise for unlearning (Ilharco et al., 2022) as well.

## B. Additional Experimental Details

### B.1. Code

Because we evaluate multiple methods on a wide range of tasks, training on $> 20$ datasets, we defer all the details on the prompts, exact dataset format, etc. to our anonymized code repository.

### B.2. Hyperparameter Ranges

**FFT.** We tune the learning rate in the range $5e - 7, 1e - 5$. We find that $1e - 6$ works as a good learning rate across all tasks. We use a batch size of 32.

**LoTA.** We do not tune any hyperparameters for LoTA and merely use the same hyperparameters as FFT.

**LoRA.** LoRA introduces the additional rank hyperparameter, which we tune **jointly** with the learning rate. This is the rank, which we tune in $4, 256$. Common wisdom seems to be to use larger learning rates for LoRA, so we expand the upper edge of the LoRA learning rate range to $1e - 4$ and indeed find that LoRA typically benefits from a larger learning rate. We set "lora alpha" to 16. We use LoRA on all linear layers.

**TIES-Merging.** We consider post-hoc sparsification factors of $0.1, 0.2, 0.3$; the best performance is at either 0.1 or 0.2.

### B.3. Additional Experiments

In Table 1, we present the results with sparse adaptation for 3 epochs. In Table 6, we provide the corresponding results with 1-epoch sparse adaptation.

*Table 6.* Performance comparison on single-task datasets for 1 epoch. **bold**: best method

| Method | Model | Arc | GSM8k | SQL | Summarization |
|---|---|---|---|---|---|
| LoRA | Mistral | $70.4_{0.6}$ | $46.3_{1.1}$ | $98.6_{0.1}$ | $51.8_{0.2}$ |
| LoTA (ours) | Mistral | $\mathbf{73.8_{0.7}}$ | $\mathbf{53.5_{1.0}}$ | $\mathbf{99.3_{0.1}}$ | $\mathbf{54.3_{2.5}}$ |

### B.4. Individual Task Results for Averaged Experiments

In the main body we present a number of experiments where we have to report the average performance over a number of tasks for space constraints. We now present the individual task results in Table 7.

*Table 7.* Individual task results for the "averaged" results in Table 2. **bold**: best method.

| FT Method on Task A | FT Method on Task B | Instruction Following | Arc | GSM8k | SQL |
|---|---|---|---|---|---|
| FFT | FFT | $0.45_{0.21}$ | $74.1_{.09}$ | $51.9_{0.09}$ | $98.9_{0.01}$ |
| LoTA (ours) | FFT | $\underline{11.48_{0.73}}$ | $73.3_{0.02}$ | $53.9_{0.09}$ | $98.9_{0.01}$ |
| LoTA (ours) | LoTTO (ours) | $\mathbf{15.88_{0.88}}$ | $70.3_{0.01}$ | $52.5_{0.01}$ | $98.6_{0.01}$ |

