# OpenReview forum: "Lottery Ticket Adaptation: Mitigating Destructive Interference in LLMs"
_ICML.cc/2024/Workshop/WANT — WANT@ICML 2024 Oral_

### Official Review · Reviewer_FYuF · 2024-06-13
**Lottery Ticket Adaptation: Mitigating Destructive Interference in LLMs**

**Confidence:** 3

**Summary:**

The paper introduces a novel method called Lottery Ticket Adaptation (LoTA) to address catastrophic forgetting and destructive interference when a model is finetuned on a new task. LoTA consists of three phases: mask calibration, mask extraction, and sparse adaptation. In the mask calibration step, the model is finetuned on the new task, converting the original parameters Wp to new parameters Wf. In the mask extraction phase, the difference between Wf and Wp is extracted as sparse matrix M. In the sparse adaptation phase, the extracted sparse matrix is used to create a subnetwork Wp * M. Only this subnetwork is finetuned on the new task. LoTA performs better than full fine-tuning and LoRA in mitigating destructive interference in different multi-task adaptation paradigms and also in preventing catastrophic forgetting of earlier tasks.

**Strengths:**

- LoTA is a novel contribution that is effective across three major multi-task adaptation methods: storing and loading adapters, sequential training, and model merging.
- The paper is well-written.
- The paper provides thorough explanations and results for each adaptation approach.
- Most of the claims are well-supported by extensive experiments, evaluating two models on six different tasks across three multi-task adaptation paradigms.

**Weaknesses:**

- LoTA requires extra time and resources to obtain the sparse matrix, which can limit its usability. While the paper acknowledges this issue, provides some solutions like using masks transferred from other datasets, and mentions that it will not add more than a few hours of compute time on a single GPU, it remains an open problem as the number of tasks increases.
- Results for single-task experiments (Shown in Table 1) are supported by both models LLama-3 and Mistral. It would be important to see if - - LLama-3 shows similar patterns in sequential training and model merging experiments too. This would enhance the robustness of the findings across models.
- The paper will benefit by addressing the effectiveness of LoTA on more than two task settings. For example, how well LoTA mitigates catastrophic forgetting for Task A, B when fine-tuned for task C.
- In Table 2, it would make the results more complete if combinations like LoRA-LoRA and LoRA-FFT were added to the GSM8K results.
- Including related work on lottery ticket hypotheses in the main paper would provide additional context for readers.

**Suggestions:**

- Page 1 right column, remove ‘[leftmargin=15pt]’.
- Page 3, line 113, right column. `Lottery Ticket Adaptation (LoTA). LoTA works in two phases as summarized in Figure 2: (1) mask calibration, (2) mask extraction, (3) sparse adaptation.`. → replace `two phases` with `three phases`.

---

### Official Review · Reviewer_zP6K · 2024-06-14
**LoTA offers a novel, efficient solution to mitigate destructive interference in LLMs, but needs clearer methodology and better reproducibility.**

**Confidence:** 5

**Summary:**

The authors introduce Lottery Ticket Adaptation (LoTA), a sparse adaptation method that identifies and optimizes a sparse subnetwork within a large language model (LLM). This method aims to mitigate destructive interference between tasks during multi-task adaptation. LoTA outperforms full fine-tuning (FFT) and low-rank adaptation (LoRA) in various tasks such as instruction following, reasoning, math, and summarization, preventing catastrophic forgetting and enabling efficient model merging. LoTA uses sparse task vectors, making it memory-efficient compared to other methods. However, the computational complexity introduced during mask calibration needs careful evaluation.

Sparse Adaptation: LoTA involves three phases:
1. Mask Calibration - initial full fine-tuning to identify significant weight updates.
2. Mask Extraction - derivation of a sparsity mask from the updates, isolating the sparse subnetwork relevant to the task.
3. Sparse Fine-Tuning - re-application of the sparsity mask and fine-tuning the identified subnetwork while freezing other parameters.

Finally, with sequential training and model merging, LoTA addresses challenges in sequential training by preventing catastrophic forgetting and improves model merging by using mutually sparse task vectors.

In terms of results, the author show that on instruction following, their method achieved a win rate of 19.0%, matching FFT and surpassing LoRA (15.3%). Also, the show mitigation of catastrophic forgetting, where their method maintained a higher win rate (17.7%) compared to FFT (15.2%) when adapting from instruction following to GSM8k. Last, for model merging they achieved better task-average performance compared to existing methods, with a performance of 38.5% in a merged model scenario.

**Strengths:**

* LoTA's sparse adaptation method is highly innovative and addresses critical challenges in multi-task learning and model merging.
* The paper includes extensive experiments across multiple tasks, validating the effectiveness of LoTA. The results consistently demonstrate LoTA's superior performance.
* The method's efficiency and ability to prevent catastrophic forgetting make it highly practical for real-world applications, especially in dynamic environments.

**Weaknesses:**

* The supplementary materials are lacking in detail, particularly regarding hyperparameter settings and code availability. This hinders reproducibility and understanding.
*  The LoTTO component introduces additional computational complexity, which could be a drawback for some applications. The paper should discuss potential optimizations to reduce this overhead.
* While the paper is sound, a deeper exploration of the theoretical underpinnings and potential mathematical optimizations of the sparsity mask calibration process would be beneficial. For instance, a more rigorous analysis of the sparsity patterns and their impact on model performance could provide valuable insights.

**Limitations:**

* The provided information is not sufficient for full replication of the experiments. The absence of exact code and dataset details in the supplementary materials limits the ability to reproduce the results.
* The mask calibration phase introduces significant computational overhead, which may limit the practicality of LoTA in resource-constrained environments. The additional steps required for sparse fine-tuning and iterative mask calibration can be computationally intensive.
* The paper lacks detailed explanations, particularly in the mask extraction and application processes. A more thorough breakdown of these steps, including mathematical formulations and justifications, would improve comprehension.

**Suggestions:**

* Future research should explore optimizing the sparsity mask calibration process to reduce computational overhead and enhance performance. Investigating the application of LoTA in other domains beyond language models could further demonstrate its versatility.
* Broadening the application scope to include other types of neural networks and tasks could extend LoTA's impact across different areas of machine learning.

---

### Meta-Review · Area_Chair_Bx7p · 2024-06-17

**Recommendation:** Accept (Oral)
**Confidence:** 5

**Metareview:**

The paper proposes a new sparse adaptation method that searches for sparse subnetwork within a large model. The approach outperforms both LoRA adapters and full finetuning on the selected datasets and tasks. Both reviewers acknowledge the paper's novelty and are overall positive about the work. The AC agrees and recommends for acceptance.

---

### Decision · Program_Chairs · 2024-06-17

**Decision:**

Accept (Oral)

**Comment:**

We thank the authors for their time and contribution to WANT and we are pleased to share that after the reviewing process the paper has been accepted. Congratulations! We encourage the authors to consider reviewers' feedback for the improvement of the camera-ready version. We hope to see you in person at the workshop and brainstorm on efficient training research together!